# Acceptability of Dry-Cured Belly (Pancetta) from Entire Males, Immunocastrates or Surgical Castrates: Study with Slovenian Consumers

**DOI:** 10.3390/foods8040122

**Published:** 2019-04-13

**Authors:** Marjeta Čandek-Potokar, Maja Prevolnik-Povše, Martin Škrlep, Maria Font-i-Furnols, Nina Batorek-Lukač, Kevin Kress, Volker Stefanski

**Affiliations:** 1Agricultural institute of Slovenia, Hacquetova ul. 17, 1000 Ljubljana, Slovenia; maja.prevolnik@kis.si (M.P.-P.); martin.skrlep@kis.si (M.Š.); nina.batorek@kis.si (N.B.-L.); 2University of Maribor, Faculty of Agriculture and Life Sciences, Pivola 10, Hoče, Slovenia; 3IRTA, Granja Camps i Armet, 17121 Monells, Spain; maria.font@irta.cat; 4University of Hohenheim, Garbenstr. 17, 70599 Stuttgart, Germany; kress.kevin@uni-hohenheim.de (K.K.); Volker.Stefanski@uni-hohenheim.de (V.S.)

**Keywords:** consumer, sensory acceptability, dry-cured belly, pancetta, pig, boar taint

## Abstract

Abandoning of male piglets castration in the European Union is a challenge for the pork production sector in particular for high-quality dry-cured traditional products. The information on consumer acceptability of dry-cured products from alternatives is limited, so the objective was to test the consumer acceptability of unsmoked traditional dry-cured belly (Kraška panceta) processed from three sex categories, i.e., surgical castrates (SC), entire males (EM) and immunocastrates (IC). Consumers (*n* = 331) were asked to taste dry-cured bellies from EM, IC and SC and to score the taste appreciation on a 9 cm unstructured scale. After tasting the pancetta of three sex categories, the consumers attributed the lowest acceptability scores to SC, whereas IC and EM received similar scores. Only about a quarter of consumers attributed the lowest score to EM, mainly when boar taint compounds were present. The results of this study indicate that a certain share of consumers was sensitive to taste deficiencies and that the leanness of this product is very important for consumers.

## 1. Introduction

Surgical castration of male piglets is a worldwide used practice in pig production with the main goal to prevent so-called boar taint—an unpleasant aroma and taste of pork. Boar taint has been associated with the presence of high levels of androstenone and/or skatole, two lipophilic substances that accumulate in fat tissue of uncastrated male pig [1,2]. Androstenone is a steroid that serves as a pheromone and is produced by testicular Leydig cells, whereas skatole is a product of bacterial degradation of the amino acid tryptophan in the large intestine. Hepatic metabolism of skatole is hindered by steroid hormones (including androstenone), so increased concentrations of androstenone are responsible for higher levels of skatole [3]. The levels above which the consumers can detect androstenone and skatole are considered to be between 0.5–1.0 µg/g fat and 0.20–0.25 µg/g fat for androstenone and skatole, respectively [4]. In practice surgical castration is done early in a piglet’s life; the European Union legislation allows this procedure to be done without the use of anesthesia or analgesia within the first week after birth [5]. However there is a growing public dissent concerning this method due to a negative effect on animal welfare. For this reason the European pork chain committed to voluntary stop male piglet castration without pain relieving [6]. However, a sustainable exit from piglet castration only works if unsolved issues are discussed critically and alternative solutions are evaluated. In particular, the industry is facing major challenges when fattening boars to higher age and weight at slaughter, which is problematic for high-quality traditional products, as the highest risks are associated with fat quantity and quality [7]. Boar taint is more easily perceived when fat content is high, no masking ingredients are used and the product is consumed warm [8]. Many processing technologies have been tested in order to mask boar taint; however, a recent review study indicated that in order for consumers to not detect it, processed pork would need to have androstenone levels lower than 0.4 µg per g and to be served at below 23 °C [9]. Boar taint was shown to be perceived in dry-cured products, even if these are not consumed warm [10,11]. A recent review of studies with consumers regarding boar taint and consumer acceptability [8] demonstrated a need for a better understanding of the risks related to the perception of boar taint by the consumer in the case of different product types. Moreover, there is a lack of studies comparing sensory acceptability of meat products made from entire males but also other alternative options like immunocastrated males.

The objective of the present study was to test the sensory acceptability of a traditional Slovenian product, dry-cured belly Kraška panceta with Slovenian consumers. Kraška panceta is a product protected with geographical indication (PGI) according to EU legislation. The origin of raw material is not prescribed which denotes procurement on different EU markets and no control over the sex category of pigs used. With increasing probability of meat from uncastrated male pigs on the market, it is important to verify the acceptability of this product made from meat of different alternatives, i.e., pig sex categories.

## 2. Materials and Methods

Bellies came from pigs (crosses of German Landrace sows and Pietrain boars raised on the same farm, fed the same standard commercial feed mixture) of three sex categories, i.e., entire males (EM), immunocastrates (IC) and surgical castrates (SC). Pigs derived from one slaughter batch and were of similar age (185.0 ± 3.4 days) and weight (121.1 ± 9.9 kg). IC received two applications of the vaccine IMPROVAC® (Zoetis Belgium SA, Louvain la Neuve, Belgium) according to the manufacturer‘s recommendations. Fresh bellies were submitted to processing (regular commercial production respecting food safety legislation) according to the rules of traditional Slovenian dry-cured belly product protected with geographical designation (PGI), Kraška panceta. The processing procedure consisted of seven days dry-salting, surface addition of spices (black pepper and garlic), no smoking, and air-drying for 12 weeks. Bacons (*n* = 18, six per sex group with average fresh belly leanness evaluated on a carcass cross section at the last rib being 72.8%, 68.7% and 54.4% for EM, IC and SC, respectively) were selected at the end of the processing for the sensory analysis with consumers. In the case of products from EM, dry-cured bellies were selected so as to cover a large range of boar taint levels (i.e., of androstenone and skatole levels, Table 1), whereas in the case of SC and IC the levels of androstenone and skatole in the fat tissue were below the limit of detection of the analytical method. Surveys with consumers were conducted at the main agricultural fair in Slovenia (AGRA 2018) and comprised of 331 volunteers (210 males, 121 females), visitors of the fair that agreed to taste and evaluate in parallel three slices of pancetta, one from EM, one from IC and one from SC. For that purpose, six sets of EM-IC-SC products were used (Table 1). Each triplet of products was tasted by 50 to 65 consumers. Slices were codified as A, B and C for SC, IC and EM, respectively and all three slices were given to the consumer at the same time. The order of tasting was a decision of the consumer. No personal data were asked and collected from the visitors that tasted the products. 

Each visitor was asked to evaluate the acceptability of the taste of three pancetta slices (one from EM, one from IC and one from SC) and to provide a note on a 9 cm non-structured scale anchored at two ends (from dislike extremely to like extremely). The distance from the left anchor to mark was measured. Data obtained were submitted to an analysis of variance (ANOVA), using the procedure Mixed of statistical software SAS® (SAS Institute Inc., Cary, NC, USA). The model included the fixed effects of the sex category (EM, IC and SC), set of products (1 to 6) and their interaction and gender of the consumer and a random effect of the consumer. Due to a significant interaction, the slicing by sex category and set of products were used to evaluate the effect of sex category within the set of products and vice versa. The least squares means (LSM) were calculated and compared (using a Tukey test for multiple comparisons). A threshold probability level considered for statistical significance was at *p* < 0.05. 

## 3. Results

The analysis of variance (Table 2) showed that the pig sex category had a significant effect on sensory acceptability of pancetta, and that the consumers attributed the pancetta from EM and IC with the highest average acceptability score (6.3 and 6.2, respectively), whereas SC products received a significantly (*p* < 0.0001) lower average score (5.4). There was no difference in the acceptability score between male and female consumers. 

There was a significant effect (*p* < 0.0001) of the interaction between the set of tested products and pig sex category on the acceptability score, indicating that the result (i.e., acceptability according to the pig sex category) was not the same in all sets of products. Differences in the liking score between EM, IC and SC were significant in the case of high tainted EM products (sets 3 and 4) and the untainted EM product (set 6) but not in low tainted EM products (Table 3).

Pancetta of SC received the lowest average score in four out of six sets (Table 3) and in total and was scored the lowest by 43.5% of consumers. In the absence of boar taint substances in EM (i.e., set 6), EM was liked the most and SC the least with IC taking an intermediate position. When EM had low boar taint (sets 1 and 2), the average acceptability score of EM was similar to IC and SC, whereas when EM had high boar taint (sets 3 and 4), IC had the highest acceptability. Overall, there was 23.3% of consumers who gave the lowest score to pancetta from EM. When androstenone level was about 0.5 ppm (sets 1 and 2), the lowest score was given by 22.8% of consumers, when androstenone level was very high (sets 3, 4 and 5), the lowest score was given by 27.2% of consumers. In the case of EM pancetta with boar taint compounds below detection limit (set 6), there was still 11.3% of consumers which gave the lowest score to EM pancetta. Figure 1 shows a decrease of liking of EM pancetta with an increasing level of boar taint substances, whereas just the opposite trend can be observed for SC and IC (increased liking of IC and SC products with increasing boar taint in EM products).

## 4. Discussion

In the present study, the slices of products were coded A, B and C for SC, IC and EM, respectively, which could bias the results as a serial position has been shown to affect preferences (large primacy effect, i.e., advantage of the first) [12]. However, the slices were presented together, the consumers were free to choose the order of their tasting (not recorded) and SC, which were coded A, were the least appreciated. The overall lowest consumer sensory appreciation of dry-cured belly products from SC was unexpected, in particular because boar taint substances were present in most of the EM products and in some of them at very high levels. It can be speculated that the consumers, despite being asked to score the taste, were evaluating the overall acceptability of the product. The context in which the sensory properties are perceived alter the perception [13]. Implicit sensory experience (i.e., the ‘taste’ of a food) represents only part of the eating experience; visual and haptic perception reveal the strongest correlations with the overall experience [14]. It can be assumed that they were biased by visual appearance, most likely the leanness with EM having the most lean and SC the fattiest bellies with IC in the middle but closer to EM (72.8, 54.4 and 68.7 lean %, respectively). This is also substantiated by the fact, that SC was scored the lowest in majority of the product sets and that IC products were scored the highest (and higher than EM) when boar taint substances were at a high level in the EM product. The maximum difference between scores given to SC, IC and EM were obtained in products of set 6, thus, without taint in EM. However, despite this generally best appreciation observed for EM products, the consumers were sensible to taste deficiencies. Firstly, about a quarter of consumers gave the lowest score to the EM product. Secondly, when EM had low boar taint, it was more similar to IC and SC, probably because consumers were not completely satisfied with the taste of EM although not enough dissatisfied to show a preference for SC or IC. However, when EM had high boar taint, IC had higher acceptability than EM, probably because the consumers did not like the EM taste and choosing between IC and SC, the first were less fatty so were preferred. This is demonstrated in Figure 1, where a downward trend could be observed for EM products, and an upward trend for products of IC and SC, which confirms a reduced appreciation in the case of EM products with boar taint. The review carried out by Font-i-Furnols in 2012 [8] showed that in two studies, the acceptability of EM pork depended on the level of androstenone or boar taint, but in two studies, the bacon from EM was equally accepted as those from other sexes (IC not included). Contrary as in the present study, Spanish consumers scored the loins from EM lower than those from SC and IC, even when the levels of androstenone were low [15]. However, in this study with loins, consumers did not see the fatness of the product, which could influence a lot the appreciation of the meat by consumers [16] and may explain the results of the present study. The literature dealing with the effect of processing technologies on boar tainted meat generally shows that dry-curing does not eliminate its perception in the products [10,11]. It is also considered that boar taint substances are not degraded or lost during the long dry-curing process [17], although in one of our studies we determined some reduction of the concentration of boar taint substances with very long maturing time in dry-cured ham [18]. However, in the case of pancetta, the process is much shorter. It has also been suggested that the curing process leads to a formation of aromatic compounds that are able to mask boar taint [19]. The added spices, garlic and black pepper could also contribute to the masking of boar taint [20]. It is also possible that high leanness of pancetta from EM used in the present study could partly explain why the consumers did not perceive taste deficiencies even more as boar taint is more easily perceived when fat content is high [8]. This investigation shows that boar taint levels cannot solely explain taste acceptability of pancetta, which corroborates with recently published meta-analysis [21] indicating that uncastrated males (EM) apart from boar taint, may present reduced meat quality and are more prone to oxidation [21,22]. Despite some methodological limitations due to the study design, this investigation provides an insight to the sensory acceptability of dry-cured belly by consumers in relation to the sex category as characterized by leanness and boar taint level.

## Figures and Tables

**Figure 1 foods-08-00122-f001:**
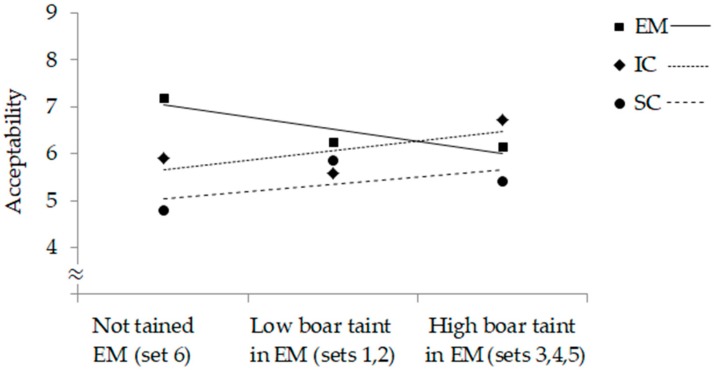
Change of consumer acceptability of pancetta according to the level of boar taint in EM. EM = entire males, IC = immunocastrates and SC = surgical castrates.

**Table 1 foods-08-00122-t001:** Level of boar taint substances in the fat tissue of entire males (EM).

	Set					
	1	2	3	4	5	6
Boar Taint Substance	Low Boar Taint		High Boar Taint			No tainT
**Androstenone ^1^**	0.55	0.45	19.5	3.07	9.10	b.d.
**Skatole ^1^**	0.03	b.d.	0.19	0.03	0.20	b.d.

**^1^** μg per g liquid fat; b.d. = below the limit of detection.

**Table 2 foods-08-00122-t002:** Analysis of variance for the pancetta sensory acceptability score.

Residual	Sex Category			Set of Products						Interaction Sex × Set	Consumer Gender	
	***p*-value**											
2.3	<0.0001			0.4308						<0.0001	0.8334	
	**LSM**											
	**EM**	**IC**	**SC**	**1**	**2**	**3**	**4**	**5**	**6**		**male**	**female**
	6.3 ^b^	6.2 ^b^	5.4 ^a^	5.9	5.9	6.4	5.9	6.0	6.0		6.0	6.0

EM = entire males, IC = immunocastrates, SC = surgical castrates; 1 to 6 sets of belly with different boar taint levels as defined in Table 1. Least squares mean (LSM) values with a different letter are statistically different at *p* < 0.05.

**Table 3 foods-08-00122-t003:** Least squares mean (LSM) for sensory acceptability according to the sex category and set of products.

	Set of products						
**Sex Category**	**1**	**2**	**3**	**4**	**5**	**6**	***p*-value**
**EM**	6.4	6.1	6.3 ^AB^	5.8 ^AB^	6.2	7.2 ^C^	0.1028
**IC**	5.4 ^a^	5.8 ^ab^	7.1 ^B,c^	6.9 ^B,bc^	6.3 ^ab^	5.9 ^B,ab^	0.0014
**SC**	5.9	5.8	5.7 ^A^	4.9 ^A^	5.5	4.8 ^A^	0.0765
***p*-value**	0.0858	0.6956	0.0074	0.0003	0.1014	<0.0001	

EM = entire males, IC = immunocastrates, SC = surgical castrates. Significant (*p* < 0.05) differences in least squares mean (LSM) values are assigned different letters (between sex categories’ uppercase letters and between sets of products’ lowercase letters).

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
