# Peer review of "Acceptability of Dry-Cured Belly (Pancetta) from Entire Males, Immunocastrates or Surgical Castrates: Study with Slovenian Consumers"

_foods, 2019, doi:10.3390/foods8040122_

Round 1
Reviewer 1 Report
This research evaluated the sensory acceptability of Slovenian consumers towards dry-cured pork belly from pigs subjected to three procedures (surgical castration, no castration, and immune-castration). The paper is well-written; however, it has major drawbacks that need to be addressed. A major concern is the lack of compositional measurements of the meat products. It seems that, from the discussion, fat was a major factor for the discrimination of the samples, and boar taint did not have the expected results because of these confounded effects. Therefore, the castration procedure may have created other physiological changes in the pig that could have resulted in changes in the meat composition. More specific comments are below:
Introduction
The introduction should include brief information regarding the effects of castration on the androstenone levels in pigs, and how this process can affect the sensory perception of the meat.
Materials and Methods
Lines 59-62: More physiological information about the pigs and the carcasses is needed (for example weight, age, fat content). It is important to know these parameters since they can affect the meat sensory acceptability.
Line 74-75: The labelling of the products may have created biases in the responses. It is a common practice to use 3- or 4-digits random codes labels to avoid any psychological effect. Besides, the order effect has been proved to affect the sensory responses. Ideally, the products should have been tested using a randomized balance order (ABC, ACB, BAC, BCA, CAB, CBA), in which each combination has an equal or similar number of consumers. The authors should address these issues and limitations of the experimental design.
Table1: What is the basis of selecting this range of boar taint in the EM treatment? Is this the natural occurrence (in terms of distribution) of the boar taint in pigs? More explanation is needed.
Line 82-83: Authors need to justify the selection of the scale. A 9-cm scale with anchors (repulsive to very tasty) seems to be unbalanced. “Repulsive” is a very negative term and this may not have an equivalent and opposite value compared to “very tasty”. For example, the 9-point hedonic scale (from “dislike extremely” to “like extremely”) has a balanced structure which it is useful for the statistical analysis.
Line 92: How many males and females participated in the tasting? This information should be stated in the materials and methods.
Figure 1: Since the gender effect (from consumers) was not significant, I recommend removing this graph and put, instead, a table of the ANOVA (Table 2) effects results (F and P values) including the gender of consumers as a treatment effect.
Table 3: From these results, the different levels of boar taint did not affect the sensory acceptability of the dry-cured meat as the EM scores varied from 5.9 to 7.2 for all products in the set. Therefore, the differences in the acceptability of the samples (EM, IC, and SC) may have been generated by other physiological changes in the pigs. This needs to be further discussed as the boar taint levels cannot solely explain why the samples were different in taste acceptability. On the other hand, the sets of products are different for IC. There is not a clear explanation of why these sets are different since they are originated from the same batch of pigs. And, if this is the effect of variability, this did not affect the EM and SC. Therefore, changes in the acceptability of the sets are procedure (castration) dependant.
Figure 2: This figure shows the comparison of three treatments according to the boar taint levels. However, in the materials and methods, it is mentioned that SC and IC did not show detectable levels of boar taint. Therefore, this comparison may not reflect the expected results of this study. It is true that mean acceptability of EM decreased (but it is not significant) with increasing boar taint levels. However, there is nothing that can be said with respect to IC and EM since the boar taint level was not measured (or at least not detected) for these meats.
Author Response
Dear Reviewer 1,
Thank you for your useful comments and suggestions. In the text, the modifications are highlighted in yellow. Below please also find our answers:
Comments and Suggestions for Authors: This research evaluated the sensory acceptability of Slovenian consumers towards dry-cured pork belly from pigs subjected to three procedures (surgical castration, no castration, and immune-castration). The paper is well-written; however, it has major drawbacks that need to be addressed. A major concern is the lack of compositional measurements of the meat products. It seems that, from the discussion, fat was a major factor for the discrimination of the samples, and boar taint did not have the expected results because of these confounded effects. Therefore, the castration procedure may have created other physiological changes in the pig that could have resulted in changes in the meat composition. More specific comments are below:.
A: With regard to the major concern - the lack of compositional measurements of the meat products -we would like to mention that the indication about lean meat content was provided in discussion – and is now given also in material and methods section (line 75). It refers to the meat cut prior to entering the processing.
Pancetta is a product that is not homogenous (like e.g. salami) so even the consumers getting the same pancetta can get compositionally (lean:meat ratio) different slice.
We agree (and discussed) that likely the consumers didn’t evaluate only taste acceptability (as asked) but rather the overall acceptability (visual and tasting). The fact that even high values of boar taint compounds present in the product were not strongly disliked is important information for meat industry.
Introduction
The introduction should include brief information regarding the effects of castration on the androstenone levels in pigs, and how this process can affect the sensory perception of the meat.
A: Information added, see lines 33-39
Materials and Methods
Lines 59-62: More physiological information about the pigs and the carcasses is needed (for example weight, age, fat content). It is important to know these parameters since they can affect the meat sensory acceptability.
A: Information about age and weight of pigs as well as the indicator of fresh belly lean meat content (per group) was added, see lines 68-69 and 75
Line 74-75: The labelling of the products may have created biases in the responses. It is a common practice to use 3- or 4-digits random codes labels to avoid any psychological effect. Besides, the order effect has been proved to affect the sensory responses. Ideally, the products should have been tested using a randomized balance order (ABC, ACB, BAC, BCA, CAB, CBA), in which each combination has an equal or similar number of consumers. The authors should address these issues and limitations of the experimental design.
A: We agree with the reviewer that the used experimental approach does not follow the classical approach in consumer studies (was simplified due to the conditions in which the study was performed i.e. tasting at the fair with by-passing volunteers). Although the product belonging to sex category was always codified with the same letter, the order of tasting was a choice of each person, and each volunteer tasted only one set of products (not all), so we consider a psychological bias minimal. Moreover, all three slices were presented in parallel (together). We can’t neglect though the potential effect of the order of tasting, which has been demonstrated to have a large primacy effect i.e. advantage of first (Mantonakis et al. 2009), however the order of tasting was a choice of consumer and even if potentially a majority of consumers tasted in order from A to C, the least appreciated were SC pancettas (coded A) and most appreciated EM pancettas were coded with C. We added some discussion on this point; see lines 137-141, lines 144-147.
Table1: What is the basis of selecting this range of boar taint in the EM treatment? Is this the natural occurrence (in terms of distribution) of the boar taint in pigs? More explanation is needed.
A: Yes, it is natural occurrence (compounds were determined in fresh subcutaneous fat tissue before processing). As explained (line 77) range of androstenone levels in EM went from zero (bellow detection of chemical method) to levels that are considered as sensory perception benchmark (set 1, 2), to very high levels (3, 4, 5). The lines 37-39 in introduction were added to inform on the levels considered as benchmark.
Line 82-83: Authors need to justify the selection of the scale. A 9-cm scale with anchors (repulsive to very tasty) seems to be unbalanced. “Repulsive” is a very negative term and this may not have an equivalent and opposite value compared to “very tasty”. For example, the 9-point hedonic scale (from “dislike extremely” to “like extremely”) has a balanced structure which it is useful for the statistical analysis.
A: We are sorry for our miswording with translation of the question to English. Actually it was from dislike extremely to like extremely. The text was corrected see line 92.
Line 92: How many males and females participated in the tasting? This information should be stated in the materials and methods.
A: This information (210 males and 121 females) was provided in the original version in l. 72 (now line 80-81)
Figure 1: Since the gender effect (from consumers) was not significant, I recommend removing this graph and put, instead, a table of the ANOVA (Table 2) effects results (F and P values) including the gender of consumers as a treatment effect.
A: Done as suggested; figure 1 was deleted, and Table adapted (see Table 1 in revised version). Effect of consumer’s gender was added in the model (also in material and methods, see lines 94-97).
Table 3: From these results, the different levels of boar taint did not affect the sensory acceptability of the dry-cured meat as the EM scores varied from 5.9 to 7.2 for all products in the set. Therefore, the differences in the acceptability of the samples (EM, IC, and SC) may have been generated by other physiological changes in the pigs. This needs to be further discussed as the boar taint levels cannot solely explain why the samples were different in taste acceptability. On the other hand, the sets of products are different for IC. There is not a clear explanation of why these sets are different since they are originated from the same batch of pigs. And, if this is the effect of variability, this did not affect the EM and SC. Therefore, changes in the acceptability of the sets are procedure (castration) dependant.
A: We agree. This is demonstrated by significant interaction (sex category × set of products), shown in Table 2 and described in lines 110 to 114. The pancetta as product is not homogenous (like e.g. salami) and reflects individual variability of pigs; and even the consumers getting the same pancetta can get compositionally (lean:meat ratio) different slice.
Figure 2: This figure shows the comparison of three treatments according to the boar taint levels. However, in the materials and methods, it is mentioned that SC and IC did not show detectable levels of boar taint. Therefore, this comparison may not reflect the expected results of this study. It is true that mean acceptability of EM decreased (but it is not significant) with increasing boar taint levels. However, there is nothing that can be said with respect to IC and EM since the boar taint level was not measured (or at least not detected) for these meats.
A: It seems that the figure was confusing. The x-axis explanation was not clear. There were no boar taint substances determined in SC and IC. To clarify we changed x-axis explanation to: not tainted EM (set 6); low boar taint in EM (sets 1,2); high boar taint in EM (sets 3,4,5) and in order not to be repetitive, this explanation was deleted from the title of Figure 1 (given in the title of former Figure 2).

Reviewer 2 Report
The manuscript entitled “Acceptability of Dry-Cured Belly (Pancetta) From Entire Males, Immunocastrates or Surgical Castrates: Study With Slovenian Consumers” presents interesting issues, but it requires also some corrections.
- Table 1 - The table must be able to stand alone and be interpretable without reading the rest of the manuscript, therefore all abbreviations should be explained (e.g.; EM)
- Line 79 – it should be “below the limit of detection” instead of “below method limit of detection”
- For the research that involves human subjects the rules of the Declaration of Helsinki of 1975 must be applied (even if no personal data were collected), including ethics commission approval and informed consent. Please add the information about number of ethics commission approval (refer this number).
- Figure 1 - The figure must be able to stand alone and be interpretable without reading the rest of the manuscript, therefore all abbreviations should be explained (e.g.; EM, IC, SC)
- This is a quite simple, but well planned experiment, which is well described. The manuscript is rather short but comprehensive and very clear in the message.
- The discussion section is not sufficient. Authors should relate the findings to those of similar studies and point out the differences and similarities between the studies. Authors should add more references different than their own (however, related to the study) in discussion section.
Author Response
Dear Reviewer 2,
Thank you for your useful comments and suggestions. In the text, the modifications are highlighted in yellow. Below please find also our answers:
The manuscript entitled “Acceptability of Dry-Cured Belly (Pancetta) From Entire Males, Immunocastrates or Surgical Castrates: Study With Slovenian Consumers” presents interesting issues, but it requires also some corrections.
- Table 1 - The table must be able to stand alone and be interpretable without reading the rest of the manuscript, therefore all abbreviations should be explained (e.g.; EM)
A: Abbreviations are now explained below the table
- Line 79 – it should be “below the limit of detection” instead of “below method limit of detection”
A: Corrected as advised.
- For the research that involves human subjects the rules of the Declaration of Helsinki of 1975 must be applied (even if no personal data were collected), including ethics commission approval and informed consent. Please add the information about number of ethics commission approval (refer this number).
A: Declaration of Helsinki refers to ethical principles for medical research involving human subjects, which is not the case in the present study. In the present study, we used meat products from regular meat production (i.e. respecting food safety legislation; added in the text, line 71) and participants were by passing visitors of the agro-food fair that stopped at stand and agreed to taste three products. It was an informed consent i.e. consumers were informed that they were tasting three different products from the same manufacturer, but they were not informed about the sex of pigs (to avoid the potential effect on his/her opinion). In order not to confuse the readers, we deleted the sentence “no information was provided to consumers about the nature of the trial”. Furthermore, no personal information was asked to the consumers, only the gender was recorded thus no personal data was collected that would demand special attention according to the General Data Protection Regulation (Regulation (EU) 2016/679 of the European Parliament and of the Council.
- Figure 1 - The figure must be able to stand alone and be interpretable without reading the rest of the manuscript, therefore all abbreviations should be explained (e.g.; EM, IC, SC)
A: Corrected as advised.
- This is a quite simple, but well planned experiment, which is well described. The manuscript is rather short but comprehensive and very clear in the message.
- The discussion section is not sufficient. Authors should relate the findings to those of similar studies and point out the differences and similarities between the studies. Authors should add more references different than their own (however, related to the study) in discussion section.
A: There are not many studies evaluating differences in dry-cured products from different sex categories (in particular from EM), none on pancetta which is a rather fatty product compared to dry-ham. Moreover this product is not submitted to smoking (masking agent). Comparisons of results from studies using different products have to be made with prudence, firstly as the products differ in the fat quantity (boar taint substances are lipophilic), second, processing technologies are different, and the products are eaten cold or warm (risk factor of perception is higher if products are eaten warm).
We enhanced the discussion – see lines 137-141, 144-147, 174-175, 179-181 and added 7 references.

Round 2
Reviewer 1 Report
The manuscript has significantly improved with the corrections and I would like to recommend its publication in this journal. However, I recommend adding more information in the discussion about the other physiological changes in pigs that happen after castration. Boar taint is one effect but there are other changes that can affect the meat composition.
Author Response
Dear Reviewer,
considering your last comment we have added the text to discussion (lines 180-182) which explains that uncastrated males (EM) apart from boar taint, present reduced meat quality and are more prone to oxidation. Two additional references were added.
Thank you for your help.
On behalf of authors, with kind regards, Marjeta Čandek Potokar
